# Clinical implications of APOBEC3A and 3B expression in patients with breast cancer

Yong-seok Kim[1☯], Der Sheng Sun[2☯], Jung-sook Yoon[3], Yoon Ho Ko[4], Hye Sung Won[2]*, Jeong Soo Kim[5]*

1 Department of Surgery, Uijeongbu St. Mary's Hospital, College of Medicine, The Catholic University of Korea, Seoul, Republic of Korea, 2 Division of Medical Oncology, Department of Internal Medicine, Uijeongbu St. Mary's Hospital, College of Medicine, The Catholic University of Korea, Seoul, Republic of Korea, 3 Clinical Research Laboratory, Uijeongbu St. Mary's Hospital, College of Medicine, The Catholic University of Korea, Seoul, Republic of Korea, 4 Division of Medical Oncology, Department of Internal Medicine, Eunpyeong St. Mary's Hospital, College of Medicine, The Catholic University of Korea, Seoul, Republic of Korea, 5 Department of Surgery, Seoul St. Mary's Hospital, College of Medicine, The Catholic University of Korea, Seoul, Republic of Korea

☯ These authors contributed equally to this work.
* btskim@catholic.ac.kr (JSK); woncomet@catholic.ac.kr (HSW)

## Abstract

### Background

We aimed to evaluate the expression of APOBEC3A (A3A), 3B (A3B) mRNA, and germline APOBEC3A/B deletion polymorphism in patients with breast cancers and to investigate the correlation between their expressions and clinicopathological characteristics.

### Methods

RNA and DNA samples were extracted from 138 breast cancer tissues and adjacent normal breast tissues. The levels of A3A and A3B mRNA transcripts were determined using quantitative real-time polymerase chain reaction. Insertion and deletion PCR assays were performed to detect the A3B deletion allele. The serum concentrations of soluble programmed death-ligand 1 (sPD-L1) and interferon gamma were determined using enzyme-linked immunosorbent assays.

### Results

A3B mRNA expression levels were significantly higher in triple-negative breast cancers compared to hormone receptor-positive, human epidermal growth factor receptor 2-negative breast cancers. Older age of the patient and high ki-67 expression were associated with increased expression levels of A3A and A3B mRNA. Advanced tumor stage, presence of lymph node involvement, and high histological grade were associated with increased expression levels of A3A mRNA. The APOBEC3A/B deletion allele was found in 77 (55.8%) patients. TP53 and PIK3CA mutations were detected in 62 (44.9%) and 31 (22.5%) patients, respectively. The presence of a PIK3CA mutation was associated with lower A3A mRNA expression levels. There was a weak positive relationship between A3A mRNA expression levels and serum sPD-L1 levels.

**Data Availability Statement:** All relevant data are within the manuscript and its Supporting Information files.

**Funding:** This work was supported by the the Basic Science Research Program through the National

Research Foundation of Korea (NRF) funded by the Ministry of Education (NRF-2016R1D1A1B03933245, W.H.S) and by The Catholic University of Korea, Uijeongbu St. Mary's Hospital Clinical Research Laboratory Foundation made in the program year of 2018 (W.H.S). The funders had no role in study design, data collection and analysis, decision to publish, or preparation of the manuscript

**Competing interests:** The authors have declared that no competing interests exist.

## Conclusions

There was a difference in A3B mRNA expression levels according to breast cancer subtypes, and high levels of A3A and A3B mRNA expressions were associated with an aggressive phenotype. There was a high incidence of APOBEC3A/B deletion allele. Further studies are needed to identify the clinical significance of APOBEC in Asian patients with breast cancer.

## Introduction

Genomic instability—a high frequency of mutations within the genome of cellular lineages—is known as the driving force of cancer development [1]. The mechanisms leading to genomic instability include inherited or acquired defects in pathways that maintain genomic integrity, such as DNA repair, DNA replication or cell cycle control [2]. DNA damage as a source of genomic instability can be caused by exogenous agents such as environmental chemicals and endogenous mutagenic events. Alexandrov LB et al. found that the catalogue of somatic mutations from a cancer genome bears the signatures of the mutational processes and a signature attributed to the Apolipoprotein B mRNA-editing enzyme, catalytic polypeptide-like (APOBEC) family is notably present in many cancer types [3]. APOBEC family is known to function in the innate immune system that protects against retroviruses by deaminating cytosine to uracil in single-stranded DNA. APOBEC has attracted much attention as an endogenous mutagen in various types of cancers. APOBEC can induce base substitutions in tumor genomes. Cytosine deamination by APOBEC in single-strand DNA results in an uracil residue. Excision of this uracil residue by uracil DNA glycosylase generates an abasic site, and insertion of adenine opposite the abasic site results in a C to T transition in a preferential motif [4]. Analyses using data from The Cancer Genome Atlas (TCGA) have revealed that the APOBEC mutation signature is particularly enriched in cancers of the bladder, cervix, head and neck, breast, and lung. Among the four intrinsic subtypes of breast cancer, it occurred much more frequently in the human epidermal growth factor receptor 2 (HER2) enriched type of breast cancer [5]. In addition, APOBEC mRNA levels correlated positively with the number of APOBEC mutation signature [5].

Humans have at least 11 APOBEC genes: activation-induced cytidinedeaminase, APOBEC1, APOBEC2 and APOBECs 3A–3H [6]. Although the roles of each APOBEC member are not yet established, previous studies have suggested that APOBEC3B (A3B) might be a major candidate inducing the APOBEC mutation signature in various cancers [7, 8]. In breast cancers, APOBEC3A (A3A) and A3B showed positive correlations between their expression levels and the APOBEC mutation signature [8–10].

With the expression of APOBEC, a germline copy number polymorphism involving A3A and A3B, which effectively deletes A3B, is known to have a role in carcinogenesis. Nik-Zainal S et al. reported that breast cancers in carriers of APOBEC3A/B deletion polymorphism showed more mutations of the putative APOBEC-dependent genome-wide signatures than cancers in non-carriers [11]. This deletion is more common in East Asians and case-control studies in Asian populations have found that the APOBEC3A/B deletion allele was associated with increased risk for breast cancer [12–14]. However, this finding was not reproduced in Western populations [15, 16].

Previous studies on the role of APOBEC in breast cancer have been conducted mainly in Western patients. We aimed to evaluate the expression of A3A and A3B mRNA according to

subtypes and the incidence of APOBEC3A/B deletion polymorphisms in Korean patients with breast cancers. We investigated correlations between their expression levels and clinicopathological characteristics and their impact on the immune response.

## Methods

### Patients

The patients who underwent surgery for breast cancer in our hospital between January 2013 and December 2016 were evaluated. Finally, this study included 138 cases with similar proportions of each subtype based on the clinicopathological data. Tissue samples were frozen immediately after surgery and stored at –70°C until DNA and RNA isolation. Venous blood samples were taken from patients at the time of surgery and centrifuged at 3,000 rpm for 10 min. The serum obtained was divided into aliquots and stored at –20°C. The following clinical data were collected: age, sex, body mass index, smoking and alcohol history, any family history of breast cancer, comorbidity, tumor stage, surgery type, surgery date, pathology reports, the use of chemotherapy, radiotherapy or endocrine therapy, tumor recurrence, and survival. Hormone receptor (HR) positivity was defined as ≥ 10% of tumor cells staining for estrogen receptor and/or progesterone receptor. HER2 positivity was defined as a HER2/CEP17 fluorescence in situ hybridization ratio of ≥ 2.0 and/or an immunohistochemical staining score of 3+. This study was approved by the institutional review board of the Catholic Medical Center (IRB No. UC18SEI0099) and was conducted in accordance with the Declaration of Helsinki. The written informed consent was obtained from all participants in this study.

### Quantitative real-time polymerase chain reaction (qPCR) assays for APOBEC3A and 3B mRNA

Total RNA was extracted from 138 breast cancer tissues and 10 adjacent normal breast tissues using RNeasy mini kits (Qiagen, Hilden, Germany) according to the manufacturer's instructions. The cDNA was synthesized from RNA using RevertAid First Strand cDNA Synthesis kits with a random hexamer primer (Fermentas International, Inc., Burlington, ON, Canada). For quantification of A3A and A3B mRNA, qPCR was performed with a CFX96 Real-time qPCR detection system (Bio-Rad, Hercules, CA, USA) using an iTaqUniversal SYBR Green Supermix (Bio-Rad). After an initial incubation at 95°C for 3 min, the reactions were followed by 40 cycles at 95°C for 10 sec and at 55°C for 30 sec. The primers used for the A3A and A3B genes were: A3A_F,5′–GAGAAGGGACAAGCACATGG–3′; A3A_R,5′–TGGATCCATCAAGT GTCTGG–3′; A3B_F,5′–GACCCTTTGGTCCTTCGAC–3′; and A3B_R,5′–GCACAGCCCCAGG AGAAG–3′, respectively [17]. The relative mRNA expression level was normalized to the β-actin gene expression, and then expressed as the log2 fold change of the delta-delta Ct values. To validate the primers for qPCR, we checked A3A and A3B mRNA levels in SKBR3 (A3B$^{null}$) and MCF-7 breast cancer cell line, and confirmed that A3B mRNA level was very low in SKBR3 breast cancer cell line (S1 Fig) [10].

### Genotyping assay of germline APOBEC3A/B deletion polymorphism

Genomic DNA was extracted from 138 matched normal breast tissues using DNeasy blood & tissue kits (Qiagen) according to the manufacturer's instructions. We designed PCR using oligonucleotide primer sequences as described previously [17]: Deletion_F,5′–TAGGTGCCACCC CGAT–3′; Deletion_R,5′–TTGAGCATAATCTTACTCTTGTAC–3′; Insertion1_F,5′–TTGGTG CTGCCCCCTC–3′; Insertion1_R,5′–TAGAGACTGAGGCCCAT–3′; and Insertion2_F,5′–TG TCCCTTTTCAGAGTTTGAGTA–3′; and Insertion2_R,5′–TGGAGCCAATTAATCACTTCAT–

3′. Insertion and deletion PCR reactions were performed separately. The PCR products for Deletion and Insertion2 primers were pooled and loaded into a standard 1.5% agarose gel. In addition, samples that appeared homozygous for the deletion, were subjected to amplifications with primer Insertion1. The following cycling conditions were used: 5 min at 95˚C, followed by 40 cycles at 95˚C for 1 min, 60˚C for 1 min, and 72˚C for 1 min. The final extension step was for 7 min at 72˚C.

## Mutations of TP53 and PIK3CA

The entire coding sequence of the TP53 gene (exons 2–11) was analyzed for mutations by direct sequencing of the PCR products. The PCR reactions for the TP53 gene were performed using Platinum Taq DNA polymerase (Thermo Fisher, Walthman, MA, USA), 10 mM dNTPs (Invitrogen Life Technologies, Carlsbad, CA, USA) and Betaine solution (Sigma-Aldrich, St. Louis, MO, USA) with the cDNA. After an initial incubation at 97˚C for 2 min, the reaction was followed by 35 cycles at 95˚C for 1 min, 67˚C for 30 sec, and 72˚C for 1 min. The hotspot mutations in exons 9 and 20 of PIK3CA were assessed by direct sequencing of the PCR products: E542K (exon 9), E545K (exon 9), H1047L (exon 20), and H1047R (exon 20) [18, 19]. The PCR reactions for the PIK3CA gene were using Ex-Taq (TaKaRa Bio Inc., Otsu, Shiga, Japan) with the cDNA. After an initial incubation at 95˚C for 5 min, the reaction was followed by 35 cycles at 95˚C for 1 min, 60˚C for 30 sec and 72˚C for 1 min. The final extension step was at 72˚C for 7 min. The primers used for the TP53 gene were: TP53_F,5′–GAGCCGCAGTCAGA TCCTAG–3′ and TP53_R,5′–GTCTGAGTCAGGCCCTTCTG–3′. The primers used for the PIK3CA gene were: exon 9_F,5′–TGGCCAGTACCTCATGGATTAGAA–3′; exon 9_R,5′–GA GGCCAATCTTTTACCAAGCA–3′; exon 20_F,5′–AATGCACAAAGACAAGAGAATTTGAG–3′; and exon 20_R,5′–AATTCCTATGCAATCGGTCTTTGC–3′.

## Determination of soluble programmed death-ligand 1 (sPD-L1) and interferon gamma (IFN-γ) concentrations

The serum concentrations of sPD-L1 and IFN-γ were determined using commercially available enzyme-linked immunosorbent assay kits in accordance with the manufacturers' instructions (Cloud-Clone Corp., Houston, TX, USA and R&D Systems Inc., Minneapolis, MN, USA). These assays use the quantitative sandwich enzyme immunoassay technique and serum levels were calculated according to standard curves. The minimum detectable levels of sPD-L1 and IFN-γ were 0.056 ng/mL and 8.0pg/mL, respectively.

## Statistical analysis

A3A and A3B mRNA levels are expressed as the median and interquartile range. A log power of 2 or fold changes in expression was used to generate box and whisker plots based on the median and interquartile range. Boxplots show the log2 fold change values for the relative expression of the APOBEC mRNA levels. Boxplots show the median (center bar) and the third and first quartiles (upper and lower edge of box, respectively). The Mann-Whitney nonparametric U test was used for comparisons of differences between continuous variables in two groups. The significance between three or more groups was tested using Kruskal-Wallis test. Dunn-Bonferroni post hoc test was employed to correct the significance for between-group comparisons. Categorical variables were compared using chi-squared or Fisher's exact tests. To determine the significant correlations between two continuous variables, we used Spearman's rank correlation coefficient. All statistical analyses were performed using SPSS program and $p < 0.05$ was considered significant.

## Results

### APOBEC3A and 3B mRNA expression according to breast cancer subtypes and characteristics

We measured A3A and A3B mRNA levels from 138 breast cancer samples and 10 adjacent normal breast tissues. A3A mRNA levels tended to be higher in the breast cancer tissues than in the normal breast tissues, but the differences were not statistically significant (median 0.20 vs. 0.16; p = 0.396). A3B mRNA levels in the breast cancer tissues were significantly higher than in the normal breast tissues (median 0.55 vs. 0.13; p = 0.013). The median mRNA levels for A3A was lower than that of A3B in the breast cancer tissues.

We classified breast cancers into three subtypes based on the expression of the estrogen receptor, progesterone receptor, and HER2: 1) HR-positive and HER2-negative breast cancer, 2) HER2-positive breast cancer, and 3) triple-negative breast cancer. The numbers of patients in each subtype were 56 (40.6%), 49 (35.5%), and 33 (23.9%), respectively. Among 49 patients with HER2-positive breast cancer, 34 (69.3%) patients were also HR-positive. With respect to breast cancer subtypes, A3A mRNA expression showed a tendency to be higher in HER2-positive breast cancers, and A3B mRNA expression was significantly higher in triple-negative breast cancers compared to HR-positive, HER2-negative breast cancers (p = 0.004) (Fig 1). In post hoc test, p-values between HR-positive breast cancers and HER2-positive breast cancers and between HER2-positive breast cancers and triple-negative breast cancers and between HR-positive breast cancers and triple-negative breast cancers were 0.853, 0.061, and 0.003, respectively. Table 1 shows the mRNA expression levels of A3A and A3B according to clinico-pathological characteristics of the 138 patients. Both A3A and A3B mRNA levels were higher in older patients and Ki-67 > 20%. Advanced tumor stage, presence of lymph node involvement, and high histological grade showed significant associations with higher A3A mRNA levels.

### Relationships between APOBEC3A/B deletion polymorphism and tumor characteristics

We performed genotyping assays for APOBEC3A/B deletion polymorphism (A3B deletion) from 138 normal breast tissues. The samples were classified as A3B$^{D/D}$ (two-copy deletion), A3B$^{D/I}$ (one-copy deletion), and A3B$^{I/I}$ (no deletion). Among these patients, 77 had A3B$^{D/I}$ (55.8%) and there was no instance of A3B$^{D/D}$. There was no association between A3A mRNA

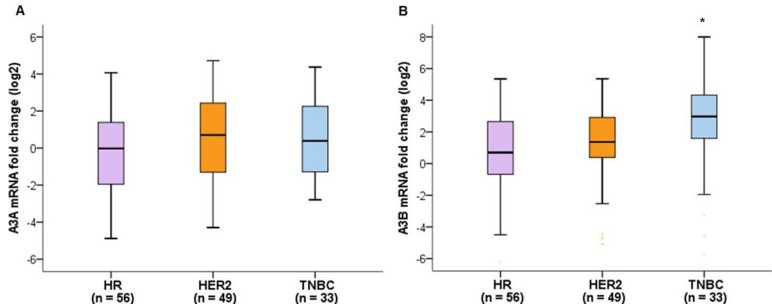

**Fig 1. Comparative quantification of APOBEC mRNA expression according to breast cancer subtypes.** (A) A3A mRNA expression levels showed a tendency to be higher in HER2-positive breast cancer (p = 0.158). (B) A3B mRNA expression levels were significantly higher in triple-negative breast cancers compared to HR-negative, HER2-positive breast cancers (p = 0.004). The significance between groups was tested using Kruskal-Wallis test followed by Dunn's multiple comparison post hoc test. Global p-values are indicated.$^{*}p < 0.05$.

**Table 1. APOBEC3A and APOBEC3B mRNA expression levels according to patient characteristics.**

| Characteristics | No. of patients (%) | APOBEC3A median (interquartile range) | p-value | APOBEC3B median (interquartile range) | p-value |
|---|---|---|---|---|---|
| Age (median: 52 years) | | | 0.012 | | 0.036 |
| ≤ 52 | 72 (52.2) | 0.16 (0.62) | | 0.37 (1.00) | |
| > 52 | 66 (47.8) | 0.32 (0.63) | | 0.92 (1.89) | |
| BMI (kg/m$^2$) | | | 0.113 | | 0.028 |
| Normal (< 23.0) | 44 (31.9) | 0.12 (0.76) | | 0.29 (0.87) | |
| Overweight (≥ 23.0) | 94 (68.1) | 0.24 (0.59) | | 0.85 (1.69) | |
| Menopausal status | | | 0.189 | | 0.852 |
| Premenopausal | 59 (43.1) | 0.16 (0.75) | | 0.54 (1.28) | |
| Postmenopausal | 78 (56.9) | 0.24 (0.59) | | 0.61 (1.68) | |
| Smoking | | | 0.933 | | 0.622 |
| Current/Ex-smoker | 14 (10.1) | 0.16 (0.66) | | 0.47 (1.81) | |
| Non-smoker | 124 (89.9) | 0.21 (0.64) | | 0.57 (1.61) | |
| Alcohol | | | 0.980 | | 0.421 |
| Yes | 10 (7.8) | 0.19 (0.55) | | 1.28 (2.6) | |
| No | 128 (92.8) | 0.21 (0.64) | | 0.54 (1.52) | |
| Family history | | | 0.483 | | 0.178 |
| Yes | 8 (5.8) | 0.72 (1.24) | | 1.38 (4.91) | |
| No | 130 (94.2) | 0.19 (0.61) | | 0.54 (1.44) | |
| Diabetes | | | 0.203 | | 0.326 |
| Yes | 22 (15.9) | 0.24 (0.74) | | 0.45 (0.83) | |
| No | 116 (84.1) | 0.19 (0.64) | | 0.62 (1.73) | |
| Hypertension | | | 0.161 | | 0.279 |
| Yes | 44 (31.9) | 0.22 (0.69) | | 0.90 (1.67) | |
| No | 94 (68.1) | 0.19 (0.60) | | 0.44 (1.36) | |
| Site | | | 0.091 | | 0.711 |
| Right | 50 (36.8) | 0.17 (0.42) | | 0.43 (1.78) | |
| Left | 86 (63.2) | 0.26 (0.73) | | 0.57 (1.40) | |
| Tumor size | | | 0.071 | | 0.901 |
| ≤ 2 cm (pT1) | 50 (36.2) | 0.18 (0.39) | | 0.41 (1.64) | |
| > 2 cm (pT2/3) | 88 (63.8) | 0.29 (0.75) | | 0.63 (1.52) | |
| No. of involved LNs | | | 0.004 | | 0.974 |
| 0 | 79 (57.2) | 0.16 (0.34) | | 0.50 (1.66) | |
| 1–3 | 36 (26.1) | 0.42 (1.01) | | 0.62 (1.19) | |
| ≥ 4 | 23 (16.7) | 0.28 (0.73) | | 0.60 (1.51) | |
| pStage | | | 0.007 | | 0.811 |
| I | 37 (26.8) | 0.10 (0.24) | | 0.37 (1.77) | |
| II | 75 (54.4) | 0.29 (0.73) | | 0.62 (1.22) | |
| III/IV | 26 (18.8) | 0.35 (1.02) | | 0.63 (1.69) | |
| Grade | | | 0.001 | | 0.067 |
| Well/ moderate | 72 (52.2) | 0.18 (0.34) | | 0.36 (1.33) | |
| Poor | 66 (47.8) | 0.39 (1.05) | | 0.71 (1.78) | |
| Ki-67 | | | 0.001 | | 0.027 |
| ≤ 20% | 57 (41.3) | 0.11 (0.46) | | 0.35 (0.92) | |
| > 20% | 81 (58.7) | 0.30 (0.81) | | 0.90 (1.83) | |
| Lymphatic invasion | | | 0.093 | | 0.142 |
| Yes | 58 (43.0) | 0.26 (0.76) | | 0.91 (1.73) | |
| No | 77 (57.0) | 0.18 (0.49) | | 0.40 (1.39) | |

*(Continued)*

**Table 1.** (Continued)

| Characteristics | No. of patients (%) | APOBEC3A median (interquartile range) | p-value | APOBEC3B median (interquartile range) | p-value |
|---|---|---|---|---|---|
| Subtype | | | 0.158 | | 0.004 |
| HR+/ HER2- | 56 (40.6) | 0.02 (0.75) | | 0.70 (1.96) | |
| HER2+ | 49 (35.5) | 0.70 (1.13) | | 1.37 (2.57) | |
| HR-/ HER2- | 33 (23.9) | 0.39 (0.97) | | 2.97 (2.90) | |

The mRNA levels are presented as the log2 fold change of the delta-delta Ct values.

The significance between groups was tested using Mann-Whitney U test and Kruskal-Wallis test followed by Dunn's multiple comparison post hoc test. Global p-values are indicated.

BMI, body mass index; No., number, LNs: lymph nodes.

levels and A3B deletions, but A3B mRNA levels showed a tendency to be lower in breast cancers with A3B deletion (p = 0.299 and 0.075, respectively) (Fig 2). There was no difference in the frequency of A3B deletion according to breast cancer subtypes. The association of A3B deletion with clinicopathological features is summarized in Table 2. We observed no significant associations between A3B deletion and any clinicopathological features of these breast cancers except for histological grade. The incidence of tumors with poor differentiation was significantly higher in breast cancers with A3B deletion.

## Analysis of TP53 and PIK3CA mutations

Among 138 patients with breast cancers, TP53 and PIK3CA mutations were detected in 62 (44.9%) and 31 (22.5%) patients, respectively. We evaluated the association between TP53 and PIK3CA mutations and A3A and A3B mRNA expression levels. The A3A and A3B mRNA expression levels did not differ significantly according to the presence of TP53 mutations. With respect to PIK3CA mutations, A3A mRNA expression levels were significantly lower in tumors carrying mutant PIK3CA than in tumors carrying wild-type PIK3CA (p = 0.038, Fig 3A), but there was no difference in A3B mRNA expression levels according to PIK3CA mutation status (p = 0.127, Fig 3B). There were no significant associations between A3B deletion and TP53, PIK3CA mutations.

## Analysis of serum concentrations of sPD-L1 and IFN-γ

We measured serum sPD-L1 and IFN-γ levels from blood samples obtained at diagnosis in 81 of the 138 patients. For sPD-L1, it was possible to measure serum concentrations in all cases,

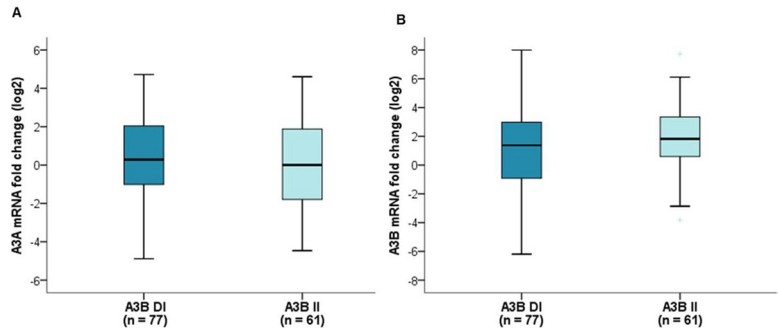

**Fig 2. Comparative quantification of APOBEC mRNA expression according to the presence of the APOBEC3A/B deletion polymorphism.** (A) There was no association between A3A mRNA expression levels and A3B deletion (p = 0.299). (B) A3B mRNA expression levels showed a tendency to be lower in breast cancers with A3B deletion (p = 0.075). A3B DI, one-copy deletion; A3B II, no deletion. Mann-Whitney U test.

**Table 2. Association of APOBEC3A/B deletion polymorphism with clinicopathological features in patients with breast cancers.**

| Characteristics | A3B$^{D/I}$ (n = 77) No.(%) | A3B$^{I/I}$ (n = 61) No.(%) | p-value |
|---|---|---|---|
| Age (year) | | | 0.696 |
| Mean±SD | 53.7±10.3 | 54.4±12.1 | |
| BMI (kg/m$^2$) | | | 0.368 |
| Normal ($<$ 23.0) | 27 (35.1) | 17 (27.9) | |
| Overweight ($\geq$ 23.0) | 50 (64.9) | 44 (72.1) | |
| Menopausal status | | | 0.668 |
| Premenopausal | 33 (42.8) | 26 (42.6) | |
| Postmenopausal | 43 (57.2) | 35 (57.4) | |
| Smoking | | | 0.214 |
| Current/Ex-smoker | 10 (13.0) | 4 (6.6) | |
| Non-smoker | 67 (87.0) | 57 (93.4) | |
| Alcohol | | | 0.348 |
| Yes | 7 (9.1) | 3 (4.9) | |
| No | 70 (90.9) | 58 (95.1) | |
| Family history | | | 0.283 |
| Yes | 3 (3.8) | 5 (8.2) | |
| No | 74 (96.2) | 56 (91.8) | |
| Diabetes | | | 0.550 |
| Yes | 11 (14.3) | 11 (18.0) | |
| No | 66 (85.7) | 50 (82.0) | |
| Hypertension | | | 0.869 |
| Yes | 25 (32.5) | 19 (31.1) | |
| No | 52 (67.5) | 42 (68.9) | |
| Site | | | 0.231 |
| Right | 30 (39.0) | 20 (32.8) | |
| Left | 47 (61.0) | 39 (63.9) | |
| Both | 0 | 2 (3.3) | |
| Tumor size | | | 0.391 |
| $\leq$ 2 cm (pT1) | 31 (40.3) | 29 (47.5) | |
| $>$ 2 cm (pT2/3) | 46 (59.7) | 32 (52.5) | |
| No. of involved LNs | | | 0.694 |
| 0 | 45 (58.4) | 34 (55.7) | |
| 1–3 | 21 (27.3) | 15 (24.6) | |
| $\geq$ 4 | 11 (14.3) | 12 (19.7) | |
| pStage | | | 0.094 |
| I | 18 (23.4) | 19 (31.1) | |
| II | 48 (62.3) | 27 (44.3) | |
| III/IV | 11 (14.3) | 15 (24.6) | |
| Grade | | | 0.014 |
| Well/ moderate | 33 (42.9) | 39 (63.9) | |
| Poor | 44 (57.1) | 22 (36.1) | |
| Ki-67 | | | 0.094 |
| $\leq$ 20% | 27 (35.1) | 30 (49.2) | |
| $>$ 20% | 50 (64.9) | 31 (50.8) | |
| Lymphatic invasion | | | 0.848 |
| Yes | 31 (41.3) | 27 (45.0) | |

(*Continued*)

**Table 2.** (Continued)

| Characteristics | A3B$^{D/I}$ (n = 77) No.(%) | A3B$^{I/I}$ (n = 61) No.(%) | p-value |
|---|---|---|---|
| No | 44 (58.7) | 33 (55.0) | |
| Subtype | | | 0.614 |
| HR+/ HER2- | 33 (42.8) | 23 (37.7) | |
| HER2+ | 28 (36.4) | 21 (34.4) | |
| HR-/ HER2- | 16 (20.8) | 17 (27.9) | |

A3B$^{D/I}$, one-copy deletion; A3B$^{I/I}$, no deletion; SD, standard deviation; BMI, body mass index; HR, hormone receptor; HER2, human epidermal growth factor receptor 2; No., number; LNs, lymph nodes.

but only 47 cases of IFN-γ were available because the serum levels were too low to be detected. The median levels of sPD-L1 and IFN-γ were 1.69 ng/mL (range, 0.174–9.757) and 1.78 pg/mL (range, 0.018–51.940), respectively. We analyzed the relationship between the serum levels of sPD-L1 and A3A, and A3B mRNA expression levels. There was a significant weak positive correlation between the serum levels of sPD-L1 and A3A mRNA expression levels (Spearman's r = 0.264, p = 0.017) (Fig 4A). The serum levels of sPD-L1 in tumors with high A3A mRNA expression levels (≥ median) and low A3A mRNA levels (< median) were 1.99 ng/mL and 1.34 ng/mL, respectively (p = 0.038) (Fig 4B). There was no significant correlation between the serum levels of sPD-L1 and A3B mRNA expression levels or between the serum levels of IFN-γ and A3A mRNA expression levels, or those of IFN-γ and A3B mRNA expression levels. There were no significant differences in serum sPD-L1 and IFN-γ levels according to A3B deletion.

## Discussion

We quantified mRNA levels of A3A and A3B in 138 breast cancer specimens and in 10 normal tissues from an adjacent area. We found that A3B mRNA expression levels were higher in breast cancer tissues than in adjacent normal breast tissues. According to breast cancer subtypes, A3B mRNA levels were significantly higher in triple-negative breast cancers compared to HR-positive, HER2-negative breast cancers. Robert SA et al. showed that APOBEC mutagenesis was the highest in HER2-enriched subtype, followed by basal-like, and luminal subtype [5]. Our studies showed that A3B mRNA levels was the highest in triple-negative breast cancers, followed by HER2-positive breast cancers, and the lowest in HR-positive, HER2-negative breast cancers. It is consistent that APOBEC mutagenesis is relatively low in HR-positive breast cancer, but there is a difference in the subtype that has the highest values. The cause of this difference is unclear, but we can consider a few things: discrepancy between immunohistochemistry-based clinical subtypes and intrinsic molecular subtypes, small sample size, and ethnic difference etc. Further studies with larger sample size of Asian population are warranted. The association of APOBEC mutagenesis with breast cancer subtypes is in line with tumor mutation burden and immunogenicity according to breast cancer subtypes. There is a higher mean mutational load in HR-negative breast cancers than in HR-positive breast cancers, and triple-negative breast cancers appear to be more immunogenic compared to other subtypes [20].

APOBEC overexpression as a potential source of genomic instability might accelerate cancer progression and be associated with poor prognostic factors. Tsuboi et al. reported that high A3B expression was associated with lymph node metastasis, lymphatic invasion, venous invasion, and high nuclear grade in Japanese patients with breast cancers [17]. Sieuwerts et al. also reported that the expression of A3B mRNA was positively correlated with nodal status, tumor size, and grade [21]. Here we found that both A3A and A3B mRNA levels were higher in

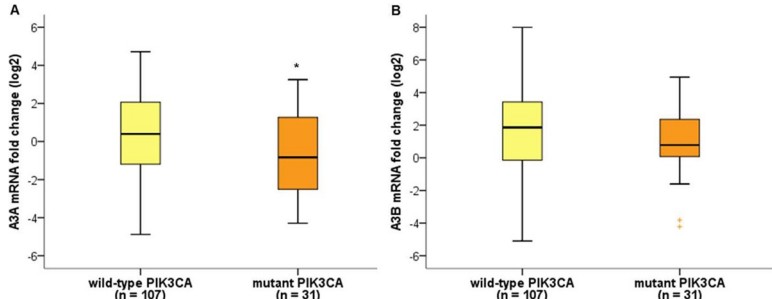

**Fig 3. Comparative quantification of APOBEC mRNA expression according to PIK3CA mutation status.** (A) A3A mRNA expression levels were significantly lower in breast cancers with PIK3CA mutations (p = 0.038). (B) There was no association between A3B mRNA expression levels and PIK3CA mutation status (p = 0.127). $^*p < 0.05$, Mann-Whitney U test.

patients with old age and Ki-67 > 20%. Advanced tumor stage, the presence of lymph node involvement, and a high histological grade showed significant associations with higher A3A mRNA levels. This suggests that APOBEC mutagenesis is linked with an aggressive phenotype in breast cancers.

TP53 and PIK3CA mutations are the most commonly altered genes in breast cancers and the association between these mutations and APOBEC expression has been studied. Based on the TCGA dataset, the overall prevalence rates of TP53 mutations are 75%, 84%, 14%, and 31% in HER2-enriched, basal-like, luminal A, and B subtypes, respectively [22]. The overall frequency of PIK3CA mutations in breast cancers has been reported as 20–40% and they are more prevalent in HR-positive and HER2-positive breast cancers than in triple-negative breast cancers [22]. The majority of PIK3CA mutations clustered in two 'hotspot' regions in exons 9 and 20, corresponding to the helical and catalytic domains of PIK3CA, respectively. Previous evidence has shown that exon 20 mutations predominate in breast cancers [18]. A previous study from Korean breast cancer cohort reported that the prevalence of TP53 and PIK3CA mutations was 47.9% and 28.5%, respectively [23]. In our study, TP53 and PIK3CA mutations were identified in 44.9% and 22.5%, respectively. All patients with PIK3CA mutations except for four were identified as having exon 20 mutations. The METABRIC and TCGA datasets showed that A3B expression was increased in tumors with TP53 mutations and the presence of a PIK3CA alteration was associated with lower A3B expression [13]. In our study, no correlations were found between A3A or A3B mRNA expression levels and TP53 mutation status.

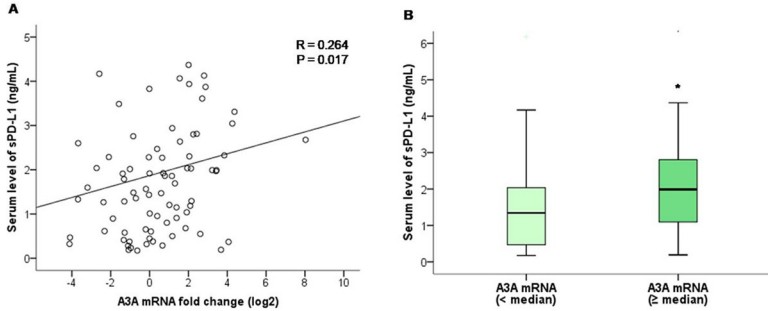

**Fig 4. Correlation of serum sPDL-1 levels and APOBEC 3A mRNA expression levels.** (A) Scatter plot graph showing a positive correlation between serum sPDL-1 levels and A3A mRNA expression levels. (B) Serum levels of sPD-L1 was higher in tumors with high A3A mRNA expression levels ($\geq$ median) than in tumors with low A3A mRNA levels (< median) (p = 0.038). $^*p < 0.05$, Mann-Whitney U test.

The presence of PIK3CA mutations was associated with low A3A mRNA expression in breast cancer. The causes and clinical significance of the relationship between PIK3CA mutations and low APOBEC expression in breast cancers are not yet clear. One possible explanation is the association with the predominant type of PIK3CA mutations in breast cancer. Henderson et al. reported that tumors with enrichment for APOBEC signature mutations were associated with PIK3CA helical domain mutations (exon 9), but not with catalytic domain mutations (exon 20) [19].

APOBEC-mediated hypermutation can induce immunogenicity, leading to increased immune-related markers. Here we evaluated serum sPD-L1 and IFN-γ levels as immune-related markers. IFN-γ is a well-known proinflammatory cytokine that is important for innate and adaptive immunity. It is produced by natural killer cells and CD4 and CD8 cytotoxic T lymphocyte effector T cells, once antigen-specific immunity develops [24]. PD-L1 is an inhibitory ligand that binds to PD-1 to suppress T cell activation. The expression of PD-L1 is induced by multiple proinflammatory molecules including IFN-γ. sPD-L1 is a soluble form of PD-L1, which can be released or shed from PD-L1-positive tumor cells or immune cells. Previous studies showed that high sPD-L1 levels were associated with a poor prognosis in various types of cancers [25, 26]. Unfortunately, the baseline levels of IFN-γ in many patients in our study were too low to make a proper analysis. However, we found that the serum levels of sPD-L1 were positively correlated with A3A mRNA expression levels. This result may suggest a possible association between APOBEC mutagenesis and tumor immunogenicity. It is obviously insufficient to draw clear conclusions. This correlation was statistically significant but weak. The roles and function of sPD-L1 in immune regulation are not fully understood. It is not yet clear whether sPD-L1 plays a significant role like membranous PD-L1, and there are questions to be answered about the clinical significance of sPD-L1. Further studies are warranted to clarify associations between APOBEC mRNA levels and tumor immunogenicity including immune cell infiltration.

Previous studies reported that germlineAPOBEC3A/B deletion polymorphism were not associated with the clinicopathological features of breast cancers, but were linked to a hypermutation phenotype [13]. The prevalence of deletion polymorphisms differs greatly between groups. There was a higher incidence (40–90%) of deletion allele in East Asian and Pacific Island populations than in African and European populations (0.9–6%) [12]. Wen et al. reported that A3B expression was significantly lower in A3B deletion carriers, but A3A expression was not observed to be significantly higher in A3B deletion carriers [27]. To detect germline APOBEC3A/B deletion polymorphism, we performed genotyping assays from 138 matched normal breast tissues. We found that 77 (55.8%) patients carried a single-copy deletion of APOBEC3A/B, and A3B mRNA expression levels showed a tendency to be lower in breast cancers with A3B deletion allele. We learned that some points should be considered when analyzing the APOBEC in Asian patients with a high frequency of A3B deletions. Recently, Starrett et al. reported that APOBEC3H haplotype I (A3H-I) associated with the APOBEC mutation signature in breast tumors lacking A3B. They suggested that combination of A3B and A3H-I explained the full APOBEC signature mutation, especially in Asia with high incidence of A3B deletion allele [28]. An analysis of A3H-I was not included in the current study, and it was a limitation of this study. Second, A3A mRNA levels can reflect values from a deletion allele as well as a non-deletion allele in case of using the common primer that used in Western studies. It is unclear whether there are differences in characteristics and levels of A3A mRNA between A3B deletion carriers and non-carriers. However, considering the possibility of the effect of A3B deletion allele on A3A mRNA, it may be more accurate analysis to measure each one separately. Further studies are necessary to clarify the effect of A3B deletion allele on

A3A mRNA levels and the best method to measure A3A mRNA levels in A3B deletion carriers.

In conclusion, A3B mRNA expression levels were significantly higher in triple-negative breast cancers compared to HR-positive, HER2-negative breast cancers. A3B and A3A mRNA expression levels were associated with aggressive phenotypes such as a high mitotic index. Tumors with PIK3CA mutations showed a relatively low A3A mRNA expression levels. There was a weak positive relationship between A3A mRNA expression and sPD-L1 as an immune-related marker. Taken together, we suggest that a combination of the mRNA expression levels of A3A and A3B might provide better clinical significance and prediction of APOBEC mutagenesis than each of these variables alone in patients with breast cancers. Further studies of including analysis of A3H-I are needed to identify more clearly the clinical significance of APOBEC in Asian patients with breast cancer.

## Supporting information

**S1 Fig. A3A and A3B mRNA expression in breast cancer cell lines.**
(TIF)

**S1 Data.**
(XLSX)

**S2 Data.**
(XLSX)

## Author Contributions

**Conceptualization:** Hye Sung Won, Jeong Soo Kim.

**Data curation:** Der Sheng Sun, Jung-sook Yoon, Yoon Ho Ko.

**Formal analysis:** Jung-sook Yoon, Yoon Ho Ko, Hye Sung Won.

**Funding acquisition:** Hye Sung Won.

**Methodology:** Jung-sook Yoon, Yoon Ho Ko.

**Resources:** Yong-seok Kim, Jeong Soo Kim.

**Supervision:** Jeong Soo Kim.

**Writing – original draft:** Yong-seok Kim, Der Sheng Sun.

**Writing – review & editing:** Hye Sung Won, Jeong Soo Kim.

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
