## [Decision Letter · Decision Letter 0]

31 Oct 2019

PONE-D-19-26098

Clinical implications of APOBEC3A and 3B expression in patients with breast cancer

PLOS ONE

Dear Dr. Won,

Thank you for submitting your manuscript to PLOS ONE. After careful consideration, we feel that it has merit but does not fully meet PLOS ONE’s publication criteria as it currently stands. Therefore, we invite you to submit a revised version of the manuscript that addresses the points raised during the review process. Please ensure all of the major comments from both reviewers are addressed, particularly providing evidence that the qPCR assays are specific.

We would appreciate receiving your revised manuscript by Dec 15 2019 11:59PM. To enhance the reproducibility of your results, we recommend that if applicable you deposit your laboratory protocols in protocols.io, where a protocol can be assigned its own identifier (DOI) such that it can be cited independently in the future. For instructions see: http://journals.plos.org/plosone/s/submission-guidelines#loc-laboratory-protocols

We look forward to receiving your revised manuscript.

Kind regards,

Elizabeth Christie

Academic Editor

PLOS ONE

Journal Requirements:

Reviewers' comments:

Reviewer's Responses to Questions

**Comments to the Author**

1. Is the manuscript technically sound, and do the data support the conclusions?

Reviewer #1: No

Reviewer #2: Partly

2. Has the statistical analysis been performed appropriately and rigorously? 

Reviewer #1: Yes

Reviewer #2: I Don't Know

3. Have the authors made all data underlying the findings in their manuscript fully available?

Reviewer #1: Yes

Reviewer #2: No

4. Is the manuscript presented in an intelligible fashion and written in standard English?

Reviewer #1: Yes

Reviewer #2: Yes

5. Review Comments to the Author

Reviewer #1: Kim and coworkers study a cohort of 138 Korean breast cancer patients and attempt to associate A3A and A3B mRNA levels with various clinical and genetic features. The results are inconclusive due to a lack of validation studies for the RTqPCR assays. Some of the rationale is also unclear.

Major:

1) Because the authors are focused an Asian populations (Korean), where an A3A-B deletion allele is common, they should also consider A3H haplotypes which Starrett and coworkers implicated in causing APOBEC signature mutations in the absence of A3B (PMID 27650891). In other words, A3H could also be clinically relevant.

2) The RTqPCR assays for A3A and A3B may not be specific due to extremely high homology between these genes. What tests have the authors done to validate these assays and ensure specificity?

3) Lines 202-207 – what are the units for A3A and A3B mRNA levels? This is unclear. All raw values should be provided in a supporting table (ie. not just medians).

4) It is hard to believe that there are no patients completely lacking A3B (ie. homozygous null) given the 77/(138x2) allele frequency of the deletion in this population. Can the deletion allele be verified using an independent assay such as with different PCR primers and sequencing?

5) The rationale for correlating serum sPD-L1 and tumor A3A and A3B mRNA levels is not explained. This analysis makes no sense at all.

Minor:

- Line 100 should also cite PMID:23389445, which is the first paper to implicate A3B in cancer

- Line 161 – dNTPs

Reviewer #2: Clinical implications of APOBEC3A and 3B expression 1 in patients with breast cancer

In this study the investigators have performed 4 main analyses of a Korean breast cancer cohort, firstly, they quantitated the mRNA levels of A3A and A3B by qRTPCR. They correlate these values with various clinical features and also the results of the subsequent analyses. Secondly, they quantified the prevalence of a germline deletion that leads to the 3’UTR of A3B being spliced onto the 3’ end of the A3A 3’ UTR, which also leads to the loss of one copy of the A3B gene in their patient cohort. Thirdly, they examined the prevalence of TP53 and PIK3CA mutations in their patient cohort by “direct” sequencing. Fourth, they correlated serum sPDL1 and IFNg protein levels with A3A and A3B mRNA expression levels. These studies found a number of statistically significant associations with either A3A or A3B levels and clinical features. Finally, they state that they performed survival analyses on the cohort by dividing the patients up based on high and low A3A and A3B expression levels. However, they found no statistically significant associations with survival. This is the first investigation of A3A and A3B in an all Korean population of breast cancer patients, and this maybe important as it is known that east Asian populations have a higher incidence of the deletion allele than western cohorts.

Of this study I have the following major concerns:

A. The methods are not described in sufficient detail to completely understand how the data was generated:

1. How were ER expression and Her2 expression measured and what were the cut-offs for being called ER positive and Her2 positive? What proportion of Her2 patients are also ER positive?

2. Does the RTPCR for A3A also amplify the deletion allele?

3. Are the mRNA levels reported in the tables deltadeltaCT values, this is not completely clear from the methods?

4. What is meant by “direct” sequencing – was this Sanger sequencing? Why was cDNA analysed and not genomic DNA? Furthermore, the exact mutations detected for each sample should be tabulated and provided as supplementary data.

5. Which statistical tests were used to generate the p values in the tables? Due to the large interquartile range I’m surprised at some of the p values. I would like to note however that as I am not familiar with all the statistical tests outlined in the methods and am unable to say whether they are the most appropriate statistical test for this data. In addition, without the full data being available (the authors only provide summary statistics) it is hard to test the veracity of the statistics.

6. What two variables are being compared to generate the statistics in fig 1? My understanding is that the K-W test that was used can demonstrate that one sample significantly dominates the other samples in the analysis, but it does not identify which sample is the dominant one and this needs to be determined using a post-hoc test?

B. The high prevalence of the deletion phenotype in this cohort (56%) makes the mRNA analysis of A3A problematic. I presume that the A3A RTPCR used will amplify mRNA from both the normal A3A allele and the deletion allele, while the A3B RTPCR would only amplify mRNA from the normal A3B allele? If this is not the case this should be clarified in the methods. If it does amplify both alleles this means that for a large number of patients the read out from the A3A assay will be measuring a combination of A3A and mutant gene expression. Thus, from the authors analysis they cannot categorically state the associations are due to A3A mRNA levels, rather than the deletion allele mRNA. While the deletion allele is identical in coding sequence the change in the 3’ UTR could change the stability of the mRNA and also the rate of translation of mRNA to protein. As the mutant allele has been associated with increased risk of breast cancer by a number of other groups the prevalence of this allele is important to the understanding of A3A biology (there is a meta-analysis of this in this publication: 10.18632/oncotarget.19400). This issue could affect a large number of the subsequent analyses that compare other clinical parameters to supposed A3A mRNA levels. Ideally the investigators would redo this analysis with an assay which can separately measure the mutant allele and normal allele.

C. The rationale for studying the link between A3A and A3B expression levels and immune related parameters such as serum IFN-gamma and sPDL1 seems tenuous. In this cohort they have not shown that the level of A3A associates with mutational burden or immune cell infiltration into the tumour. Furthermore, as they were unable to measure IFN-gamma in a large number of samples and the association of sPDL1 with A3A is marginal (R2=0.03) I think this data is not very useful/informative. Particularly as the association may or may not be confounded by the relative abundance of the deletion allele.

D. As the investigators show that the levels of A3B associate with the TNBC subtype then subsequent associations of A3B with clinical parameters need to have the caveat that the association may be due to differences in breast cancer subtype proportions within the groups studied rather than A3B directly.

E. I do not think that the survival analyses are informative and should be removed. As they only finished collecting samples from this cohort in December 2016 there is insufficient follow-up time to draw conclusions about survival. If they do want to include survival data then the Kaplan-Meier curves need to be included, and these should also state the numbers of censored events over time. In addition, it should be noted that the survival data could also be influenced by other factors such as that the population analysed contains all the different subtypes of breast cancer that would each have been treated differently (hormone therapy for ER pos, Herceptin for Her2, and chemo for TNBC). Although the numbers would be quite small they could do separate analyses for each subtype of breast cancer.

Minor points:

This cohort seems unusual with only 40% luminal breast cancer patients (most cohorts have >70% luminal subtype), the authors should comment on why this is the case. Is this normal for a Korean population or did the selection criteria skew the relative abundance of luminal patients?

The authors should also clarify whether the deletion allele is associated with any particular subtype of breast cancer (this is not completely clear from table 2 as they do not include breast cancer subtypes).

6. PLOS authors have the option to publish the peer review history of their article (what does this mean?). If published, this will include your full peer review and any attached files.

Reviewer #1: No

Reviewer #2: No

---

## [Author Response · Author response to Decision Letter 0]

23 Nov 2019

Dear Editor,

We’re very grateful for the constructive comments of the reviewers. It is clear that comments make our paper better. We revised our manuscript in response to reviewers’ comment. We have highlighted the major changes with a blue color. We hope that the revision satisfies reviewers’ requirement. Thank you very much.

Reviewer 1

1) Because the authors are focused an Asian populations (Korean), where an A3A-B deletion allele is common, they should also consider A3H haplotypes which Starrett and coworkers implicated in causing APOBEC signature mutations in the absence of A3B (PMID 27650891). In other words, A3H could also be clinically relevant.

We appreciate your good comments. New findings on APOBEC mutation signature in cancer have been reported in recent years. We planned to investigate the most studied APOBEC3A and 3B in our patients, because there was not much data on other APOBEC family including APOBEC3H in cancer. We added the A3H haplotype you mentioned as a limitation of our study in discussion.

2) The RTqPCR assays for A3A and A3B may not be specific due to extremely high homology between these genes. What tests have the authors done to validate these assays and ensure specificity?

We agree with reviewer’s opinion. We confirmed that using Primer-BLAST-NCBI. We attached the captured image of the results.

3) Lines 202-207 – what are the units for A3A and A3B mRNA levels? This is unclear. All raw values should be provided in a supporting table (ie. not just medians).

Thank you for your comments. The relative mRNA expression level was normalized to the β-actin gene expression, and then expressed as thelog2 fold change of the delta-delta Ct values. We added it in Method. We submit all raw values as supplement file (S1).

4) It is hard to believe that there are no patients completely lacking A3B (ie. homozygous null) given the 77/(138x2) allele frequency of the deletion in this population. Can the deletion allele be verified using an independent assay such as with different PCR primers and sequencing?

Yes, the deletion allele was verified using different PCR primers. We rechecked for heterozygous deletion/ insertion using insertion primer in samples that showed homozygous deletion in genotyping. (Reference: Kidd JM et al. Population stratification of a common APOBEC gene deletion polymorphism. PLoS Genet 3(4): e63)

5) The rationale for correlating serum sPD-L1 and tumor A3A and A3B mRNA levels is not explained. This analysis makes no sense at all.

Thank you for your comment. APOBEC is well known as an endogenous mutagen in various cancers. High tumor mutational burden and APOBEC mutagenesis signature are associated with immune signatures and with increased expression of immune-related genes. Therefore, we wanted to evaluate the relationships between APOBEC mRNA levels and serum levels of sPD-L1 and IFN-γ as immune-related markers. Obviously, there are better ways to evaluate cancer immunogenicity including immune cell infiltration. Unfortunately, we were not able to evaluate it properly. We described this as a limitation of our research in the discussion.

Minor

- Line 100 should also cite PMID:23389445, which is the first paper to implicate A3B in cancer.

We added that paper according to your comment. Thank you.

- Line 161 – dNTPs

We revised it. Thank you.

Reviewer 2

1. How were ER expression and Her2 expression measured and what were the cut-offs for being called ER positive and Her2 positive? What proportion of Her2 patients are also ER positive?

Thank you for your comments. We added definitions of hormone receptor positivity and HER2 positivity in method. Among 49 patients with HER2-positive breast cancer, 34 patients were also HR-positive. We added it in result.

2. Does the RTPCR for A3A also amplify the deletion allele?

With reference to other papers measuring A3A mRNA levels, we used the A3A primer that was amplified from both the normal A3A allele and the deletion allele.

3. Are the mRNA levels reported in the tables deltadeltaCT values, this is not completely clear from the methods?

Thank you for your comments. The relative mRNA expression level was normalized to the β-actin gene expression, and then expressed as the log2 fold change of the delta-delta Ct values. We added it in Method.

4. What is meant by “direct” sequencing – was this Sanger sequencing? Why was cDNA analysed and not genomic DNA? Furthermore, the exact mutations detected for each sample should be tabulated and provided as supplementary data.

To evaluate mutations in exons, cDNA was amplified by PCR with each primer of TP53 and PIK3CA, and then we did Sanger sequencing for the products. Finally, we analyzed the mutations by comparing the sequencing results with reference sequence. We submit all raw values as supplement file (S2).

5. Which statistical tests were used to generate the p values in the tables? Due to the large interquartile range I’m surprised at some of the p values. I would like to note however that as I am not familiar with all the statistical tests outlined in the methods and am unable to say whether they are the most appropriate statistical test for this data. In addition, without the full data being available (the authors only provide summary statistics) it is hard to test the veracity of the statistics.

To compare the differences of continuous variables between two or more groups, we performed Mann-Whitney U test and Kruskal-Wallis test as nonparametric tests because our data isn’t normally distributed. We added the statistical methods below the table and figure.

6. What two variables are being compared to generate the statistics in fig 1? My understanding is that the K-W test that was used can demonstrate that one sample significantly dominates the other samples in the analysis, but it does not identify which sample is the dominant one and this needs to be determined using a post-hoc test?

We appreciate your good comment. Dunn-Bonferroni post hoc testing is carried out on each pair of groups after a Kruskal-Wallis test. We added detailed information and results about this.

B. The high prevalence of the deletion phenotype in this cohort (56%) makes the mRNA analysis of A3A problematic. I presume that the A3A RTPCR used will amplify mRNA from both the normal A3A allele and the deletion allele, while the A3B RTPCR would only amplify mRNA from the normal A3B allele? If this is not the case this should be clarified in the methods. If it does amplify both alleles this means that for a large number of patients the read out from the A3A assay will be measuring a combination of A3A and mutant gene expression. Thus, from the authors analysis they cannot categorically state the associations are due to A3A mRNA levels, rather than the deletion allele mRNA. While the deletion allele is identical in coding sequence the change in the 3’ UTR could change the stability of the mRNA and also the rate of translation of mRNA to protein. As the mutant allele has been associated with increased risk of breast cancer by a number of other groups the prevalence of this allele is important to the understanding of A3A biology (there is a meta-analysis of this in this publication: 10.18632/oncotarget.19400). This issue could affect a large number of the subsequent analyses that compare other clinical parameters to supposed A3A mRNA levels. Ideally the investigators would redo this analysis with an assay which can separately measure the mutant allele and normal allele.

We appreciate your comment and understand your suggestion. We used the primer mentioned in previous studies. It measured the A3A mRNA levels from both deletion allele and non-deletion allele. As you mentioned, there may be differences between A3A mRNA from deletion allele and A3A mRNA from non-deletion allele. Some study reported that APOBEC3B expression was significantly lower in A3B deletion carriers, but APOBEC3A expression was not observed to be significantly higher in A3B deletion carriers. Some study mentioned that A3A mRNA from deletion allele was more stable, resulting in higher intracellular A3A levels. We think that further studies are needed to clarify the effect of A3B deletion allele on A3A mRNA levels and the best method to measure A3A mRNA levels in A3B deletion carriers. We added this in discussion. Thank you for your important advice.

C. The rationale for studying the link between A3A and A3B expression levels and immune related parameters such as serum IFN-gamma and sPDL1 seems tenuous. In this cohort they have not shown that the level of A3A associates with mutational burden or immune cell infiltration into the tumour. Furthermore, as they were unable to measure IFN-gamma in a large number of samples and the association of sPDL1 with A3A is marginal (R2=0.03) I think this data is not very useful/informative. Particularly as the association may or may not be confounded by the relative abundance of the deletion allele.

Thank you for your comment. We agree with your opinion. We describe this as a limitation of our research in the discussion.

D. As the investigators show that the levels of A3B associate with the TNBC subtype then subsequent associations of A3B with clinical parameters need to have the caveat that the association may be due to differences in breast cancer subtype proportions within the groups studied rather than A3B directly.

We understand your concerns. However, correlation with clinical parameters showed more apparent in A3A than in A3B. Actually, there is no big difference in A3A mRNA levels between subtypes. Therefore, we think that it’s hard to say that associations of A3B with clinical parameters are only caused by differences in breast cancer subtype proportions.

E. I do not think that the survival analyses are informative and should be removed. As they only finished collecting samples from this cohort in December 2016 there is insufficient follow-up time to draw conclusions about survival. If they do want to include survival data then the Kaplan-Meier curves need to be included, and these should also state the numbers of censored events over time. In addition, it should be noted that the survival data could also be influenced by other factors such as that the population analysed contains all the different subtypes of breast cancer that would each have been treated differently (hormone therapy for ER pos, Herceptin for Her2, and chemo for TNBC). Although the numbers would be quite small they could do separate analyses for each subtype of breast cancer.

We agreed with your opinion. Survival analyses have some limitations due to small number of patients and short follow-up time. We decided to remove the survival data according to your recommendation.

Minor points:

This cohort seems unusual with only 40% luminal breast cancer patients (most cohorts have >70% luminal subtype), the authors should comment on why this is the case. Is this normal for a Korean population or did the selection criteria skew the relative abundance of luminal patients?

We agreed with your opinion. Luminal subtype accounts for about 60-70% of all patients with breast cancer. Based on the clinicopathological data, we chose samples with similar proportions for each subtype to prevent the number of one subtype from becoming too small. We added this comments in method. Thank you.

The authors should also clarify whether the deletion allele is associated with any particular subtype of breast cancer (this is not completely clear from table 2 as they do not include breast cancer subtypes).

We described that in result (Line 241-242):There was no difference in the frequency of A3B deletion according to breast cancer subtypes.

---

## [Decision Letter · Decision Letter 1]

20 Jan 2020

PONE-D-19-26098R1

Clinical implications of APOBEC3A and 3B expression in patients with breast cancer

PLOS ONE

Dear Dr. Won,

Thank you for submitting your manuscript to PLOS ONE. After careful consideration, we feel that it has merit but does not fully meet PLOS ONE’s publication criteria as it currently stands. Therefore, we invite you to submit a revised version of the manuscript that addresses the points raised during the review process.

Please address all of the reviewers comments - both the new issues that have been identified and previous comments from Reviewer 1.

We would appreciate receiving your revised manuscript by Mar 05 2020 11:59PM. To enhance the reproducibility of your results, we recommend that if applicable you deposit your laboratory protocols in protocols.io, where a protocol can be assigned its own identifier (DOI) such that it can be cited independently in the future. For instructions see: http://journals.plos.org/plosone/s/submission-guidelines#loc-laboratory-protocols

We look forward to receiving your revised manuscript.

Kind regards,

Elizabeth Christie

Academic Editor

PLOS ONE

Reviewers' comments:

Reviewer's Responses to Questions

**Comments to the Author**

1. If the authors have adequately addressed your comments raised in a previous round of review and you feel that this manuscript is now acceptable for publication, you may indicate that here to bypass the “Comments to the Author” section, enter your conflict of interest statement in the “Confidential to Editor” section, and submit your "Accept" recommendation.

Reviewer #1: (No Response)

Reviewer #2: (No Response)

2. Is the manuscript technically sound, and do the data support the conclusions?

Reviewer #1: No

Reviewer #2: Partly

3. Has the statistical analysis been performed appropriately and rigorously? 

Reviewer #1: N/A

Reviewer #2: I Don't Know

4. Have the authors made all data underlying the findings in their manuscript fully available?

Reviewer #1: Yes

Reviewer #2: No

5. Is the manuscript presented in an intelligible fashion and written in standard English?

Reviewer #1: Yes

Reviewer #2: No

6. Review Comments to the Author

Reviewer #1: The authors have not yet addressed my most important comments. Just because they are using published PCR assays does not mean they are excused from doing proper controls for assay specificity (prior work could be incorrect). For instance, both RTqPCR and PCR assay specificity could be validated using breast cancer cell lines with known APOBEC genotypes. SKBR3, for instance, is A3B null (PMID 23389445). MCF10A is wildtype (as far as anyone knows) and A3B is inducible with PKC agonists and A3A with interferon-alpha (PMID 26420215).

Reviewer #2: Clinical implications of APOBEC3A and 3B expression in patients with breast cancer

(Revised version)

In this study the investigators have performed 4 main analyses of a Korean breast cancer cohort, firstly, they quantitated the mRNA levels of A3A and A3B by qRTPCR. They correlate these values with various clinical features and also the results of the subsequent analyses. Secondly, they quantified the prevalence of a germline deletion that leads to the 3’UTR of A3B being spliced onto the 3’ end of the A3A 3’ UTR, which also leads to the loss of one copy of the A3B gene in their patient cohort. Thirdly, they examined the prevalence of TP53 and PIK3CA mutations in their patient cohort by Sanger sequencing. Fourth, they correlated serum sPDL1 and IFNg protein levels with A3A and A3B mRNA expression levels. These studies found a number of statistically significant associations with either A3A or A3B levels and clinical features. This is the first investigation of A3A and A3B in an all Korean population of breast cancer patients, and this maybe important as it is known that east Asian populations have a higher incidence of the deletion allele than western cohorts.

The authors have addressed and clarified a number of my issues with the original manuscript. With the exception that the raw data they provided, as this was not annotated with information such as breast cancer subtype it was not very helpful. In addition, supplementary table 2 only roughly summarises the mutations discovered and does not list the exact mutations. Unfortunately, upon re-reading the manuscript a number of additional issues came up these are listed below.

Major points:

1. In the discussion they state they use matched adjacent normal tissue in their analyses but in the results section they state only 10 normal tissue samples were used and from the supplementary table 1 it appears they have used an average normal value that is not matched to the patient tumour samples. This needs to be clarified. In addition, did they include a similar distribution of mutant allele carriers in the normal tissue as in the tumour samples? Does the deletion allele affect expression of A3A and A3B in normal tissue? If this could be a confounding factor it should be discussed.

2. In both the results and discussion, it is stated that A3A and A3B are higher in younger patients, however, the results that are in table 1 show the opposite, with higher median levels of both A3A and A3B in the older (>52) patients. This needs to be clarified and corrected.

Minor points:

1. The full breast cancer subtype data and associations with A3A, A3B and the mutation allele should be included in Tables 1 and 2 for completeness.

2. Were outliers included or excluded in the generation of the line of best fit for the R-squared analysis of sPDL1 and A3A levels? While there is a significant association between sPDL1 and A3A levels the R squared value is very low, this should be noted in the results or discussion.

3. In the introduction it is stated that previous studies have shown that the Apobec mutation profile is highest in the Her2 subtype, this should be commented on in relation to the results from this study in the discussion.

4. A number of times it is stated that A3B levels are significantly higher in TNBC, this should be qualified with the fact that this is only in comparison to HR positive cancers not Her2 positive.

5. The number of Her2 positive patients in Table 2 equals 48 whereas the text states there were 49 Her2 patients.

6. The initial manuscript was written with good English however a number of the revisions/edits added to the manuscript following the initial review need to be checked for grammar.

7. PLOS authors have the option to publish the peer review history of their article (what does this mean?). If published, this will include your full peer review and any attached files.

Reviewer #1: No

Reviewer #2: No

---

## [Author Response · Author response to Decision Letter 1]

5 Feb 2020

Dear Editor,

We once again appreciate good comments of the reviewers and it is definitely helpful. We revised our manuscript in response to reviewers’ comments. We have highlighted the major changes with a blue color. We hope that the revision satisfies reviewers’ requirement. Thank you very much.

Reviewer 1 

1) The authors have not yet addressed my most important comments. Just because they are using published PCR assays does not mean they are excused from doing proper controls for assay specificity (prior work could be incorrect). For instance, both RTqPCR and PCR assay specificity could be validated using breast cancer cell lines with known APOBEC genotypes. SKBR3, for instance, is A3B null (PMID 23389445). MCF10A is wildtype (as far as anyone knows) and A3B is inducible with PKC agonists and A3A with interferon-alpha (PMID 26420215).

We appreciate your good comment. According to your recommendation, we checked the A3A and A3B mRNA expression levels in SKBR3 and MCF7 breast cancer cell lines. We confirmed that A3B mRNA level in SKBR3 (A3B null type) cells was relatively very low (undetectable level). We submited the result as a supporting figure (S1 Fig). We described that in method.

Reviewer 2

Major points:

1) In the discussion they state they use matched adjacent normal tissue in their analyses but in the results section they state only 10 normal tissue samples were used and from the supplementary table 1 it appears they have used an average normal value that is not matched to the patient tumour samples. This needs to be clarified. In addition, did they include a similar distribution of mutant allele carriers in the normal tissue as in the tumour samples? Does the deletion allele affect expression of A3A and A3B in normal tissue? If this could be a confounding factor it should be discussed.

Thank you for your comment. First, to make sure that A3A and A3B mRNA levels are increased in tumor tissues rather than normal breast tissues, we got the mRNA samples from 138 breast tumor tissues and 10 adjacent normal breast tissues of 138 patients, and compared the mRNA levels between tumor tissues and normal tissues. Second, in the case of genotyping assay of germline APOBEC3A/B deletion polymorphism, it is not a somatic mutation but a germline mutation. Therefore, we got the genomic DNA from 138 matched normal breast tissues, not tumor tissues. We’re sorry for causing confusion. To make this clear, we added the number of cases in each method, results and discussion.

2) In both the results and discussion, it is stated that A3A and A3B are higher in younger patients, however, the results that are in table 1 show the opposite, with higher median levels of both A3A and A3B in the older (>52) patients. This needs to be clarified and corrected.

Thank you for finding our mistake. We confirmed that the result in Table was correct. We revised the sentence in results and discussion.

Minor points:

1) The full breast cancer subtype data and associations with A3A, A3B and the mutation allele should be included in Tables 1 and 2 for completeness.

We appreciate your good comment. We added and revised the breast cancer subtype data in Table 1 and 2 according to your recommendation.

2) Were outliers included or excluded in the generation of the line of best fit for the R-squared analysis of sPDL1 and A3A levels? While there is a significant association between sPDL1 and A3A levels the R squared value is very low, this should be noted in the results or discussion.

We appreciate your good comment. We included all the values in correlation analysis. We changed coefficient of determination (r2) to correlation coefficient (r). The data has showed that there is a significant weak positive correlation (r = 0.264, p = 0.017). We mentioned this in results and discussion.

3) In the introduction it is stated that previous studies have shown that the Apobec mutation profile is highest in the Her2 subtype, this should be commented on in relation to the results from this study in the discussion.

Thank you for your comment. We added this in the discussion according to your recommendation.

4) A number of times it is stated that A3B levels are significantly higher in TNBC, this should be qualified with the fact that this is only in comparison to HR positive cancers not Her2 positive.

Thank you for your comment. We agreed with your opinion. We revised the sentence in results and discussion.

5) The number of Her2 positive patients in Table 2 equals 48 whereas the text states there were 49 Her2 patients.

Thank you for your comment. We revised the breast cancer subtype data according to your previous recommendation (No.1).

6) The initial manuscript was written with good English however a number of the revisions/edits added to the manuscript following the initial review need to be checked for grammar.

 � We agree with your opinion. We will correct the final version again via professional editing service web site (Online English -http://www.oleng.com.au/).

---

## [Decision Letter · Decision Letter 2]

26 Feb 2020

Clinical implications of APOBEC3A and 3B expression in patients with breast cancer

PONE-D-19-26098R2

Dear Dr. Won,

We are pleased to inform you that your manuscript has been judged scientifically suitable for publication and will be formally accepted for publication once it complies with all outstanding technical requirements.

With kind regards,

Elizabeth Christie

Academic Editor

PLOS ONE

Additional Editor Comments (optional):

Reviewers' comments:

Reviewer's Responses to Questions

**Comments to the Author**

1. If the authors have adequately addressed your comments raised in a previous round of review and you feel that this manuscript is now acceptable for publication, you may indicate that here to bypass the “Comments to the Author” section, enter your conflict of interest statement in the “Confidential to Editor” section, and submit your "Accept" recommendation.

Reviewer #1: All comments have been addressed

Reviewer #2: All comments have been addressed

2. Is the manuscript technically sound, and do the data support the conclusions?

Reviewer #1: Yes

Reviewer #2: Yes

3. Has the statistical analysis been performed appropriately and rigorously? 

Reviewer #1: Yes

Reviewer #2: Yes

4. Have the authors made all data underlying the findings in their manuscript fully available?

Reviewer #1: Yes

Reviewer #2: Yes

5. Is the manuscript presented in an intelligible fashion and written in standard English?

Reviewer #1: Yes

Reviewer #2: Yes

6. Review Comments to the Author

Reviewer #1: The authors have addressed my comments; however, fig 4 should be expanded to show both A3A and A3B correlations with PDL1 (even if the A3B results are negative).

Reviewer #2: The authors have now addressed my concerns and i'm happy for their paper to be published in PLOSone.

7. PLOS authors have the option to publish the peer review history of their article (what does this mean?). If published, this will include your full peer review and any attached files.

Reviewer #1: No

Reviewer #2: No

---

## [Editor Report · Acceptance letter]

3 Mar 2020

PONE-D-19-26098R2 

Clinical implications of APOBEC3A and 3B expression in patients with breast cancer 

Dear Dr. Won:

I am pleased to inform you that your manuscript has been deemed suitable for publication in PLOS ONE. Congratulations! Your manuscript is now with our production department. 

With kind regards,

on behalf of

Dr. Elizabeth Christie 

Academic Editor

PLOS ONE